# A Novel Thienopyrimidine Analog, TPH104, Mediates Immunogenic Cell Death in Triple-Negative Breast Cancer Cells

**DOI:** 10.3390/cancers13081954

**Published:** 2021-04-18

**Authors:** Diwakar Bastihalli Tukaramrao, Saloni Malla, Siddharth Saraiya, Ross Allen Hanely, Aniruddha Ray, Shikha Kumari, Dayanidhi Raman, Amit K. Tiwari

**Affiliations:** 1Department of Pharmacology and Experimental Therapeutics, College of Pharmacy & Pharmaceutical Sciences, University of Toledo, Toledo, OH 43614, USA; diwakarbt02@gmail.com (D.B.T.); saloni.malla@rockets.utoledo.edu (S.M.); ross.hanely@utoledo.edu (R.A.H.); shikha.pharma08@gmail.com (S.K.); 2Department of Radiation Oncology, College of Medicine, University of Toledo, Toledo, OH 43614, USA; saraiya.s@gmail.com; 3Department of Physics, College of Natural Sciences, University of Toledo, Toledo, OH 43614, USA; aniruddha.ray@utoledo.edu; 4Department of Cancer Biology, College of Medicine, University of Toledo, Toledo, OH 43614, USA; dayanidhi.raman@utoledo.edu

**Keywords:** triple-negative breast cancer, thienopyrimidine analog, immunogenic cell death, non-apoptotic cell death, dendritic cells, TNF-α

## Abstract

**Simple Summary:**

Triple-negative breast cancer (TNBC) is the most lethal and aggressive subtype of breast cancer that lacks an estrogen receptor, the progesterone receptor and the human epidermal growth factor receptor 2 (HER2), making it unsuitable for hormonal- or HER2-based therapy. TNBC is known for its higher relapse rate, poorer prognosis and higher rate of metastasis compared to non-TNBC because although patients initially respond to chemotherapy that kills cancer cells through a form of programmed cell death called apoptosis, they later develop chemoresistance and stop responding to the treatment, accounting for one fourth of all breast cancer deaths. In this study, we report a novel compound, TPH104, that elicits a unique, non-apoptotic cell death in TNBC cells. Upon treatment with TPH104, TNBC cells swell and burst, releasing immunogenic markers that alert and activate the immune system to further recognize and attack the neighboring breast cancer cells.

**Abstract:**

Enhancing the tumor immunogenic microenvironment has been suggested to circumvent triple-negative breast cancer (TNBC) resistance and increase the efficacy of conventional chemotherapy. Here, we report a novel chemotherapeutic compound, TPH104, which induces immunogenic cell death in the TNBC cell line MDA-MB-231, by increasing the stimulatory capacity of dendritic cells (DCs), with an IC_50_ value of 140 nM. TPH104 (5 µM) significantly increased ATP levels in the supernatant and mobilized intracellular calreticulin to the plasma membrane in MDA-MB-231 cells, compared to cells incubated with the vehicle. Incubating MDA-MB-231 cells for 12 h with TPH104 (1–5 µM) significantly increased TNF-α mRNA levels. The supernatants of dying MDAMB-231 cells incubated with TPH104 increased mouse bone marrow-derived DC maturation, the expression of MHC-II and CD86 and the mRNA expression of TNF-α, IL-6 and IL-12. Overall, these results indicate that TPH104 induces immunogenic cell death in TNBC cells, in part, by activating DCs.

## 1. Introduction

The activation of the immune system in cancer plays a critical role in determining the long-term success of anticancer therapies [1,2,3,4,5,6]. The induction of immunogenic cell death (ICD) by cancer chemotherapy can activate the immune system to target cancer [1,4,7]. ICD is characterized by an alteration in the levels of calreticulin (CRT) in the plasma membrane of cancer cells [7] and the release of damage-associated molecular pattern (DAMP) molecules, such as ATP and high mobility group B1 (HMGB1), from dying cancer cells [1,8]. DAMPs activate dendritic cells (DCs), leading to the cross-presentation of tumor antigens to cytotoxic T lymphocytes (CTL) that mediates the tumor-specific immune response [7,9,10]. DCs play a critical role in shaping the antitumor immune response at the interface of the innate and adaptive immune systems [11]. Frequently, cancer cells are not recognized by sentinel immune cells (immune cloaking) or the tumor-infiltrated immune cells become quiescent or anergic through co-opting of the immune cells, thus evading the immune response. This immunoediting allows tumor cells to survive, proliferate and metastasize to distant sites [12]. Therefore, augmenting the DC response through DC-based vaccination approaches *ex vivo* [13] or by activating DCs in the tumor microenvironment (TME) by chemotherapeutic drugs (e.g., anthracyclines, cyclophosphamide and oxaliplatin) is important. The induction of ICD may become more effective approach in cancer immunotherapy [4,8,14].

The efficacy of chemotherapeutic drugs to increase tumor cell death by immune activation is primarily attributed to the rapid translocation of preformed intracellular CRT to the outer leaflet of the plasma membrane (PM) of the tumor cells [7,14]. The translocated CRT can be a ligand for phagocytic receptors or adaptor proteins known to interact with CRT in the lipid rafts of the PM [15] that facilitate uptake of fragments from dying cells by the DCs [15,16]. Currently, studies have demonstrated that the translocation of CRT to the PM requires signaling events that induce endoplasmic reticulum (ER) stress, apoptosis and a terminal translocation module to expose CRT on the cell surface [14,17]. Other important biochemical events associated with ICD include the active release of ATP and HMGB1 to the extracellular milieu [8]. Released ATP induces the recruitment of antigen-presenting cells (APCs), including DCs at the tumor site, by activating the purinergic receptors P2RX7 and P2RY2 [18]. Although the underlying mechanism of HMGB1 release from cells undergoing ICD remains to be elucidated, it has been shown that HMGB1 mediates significant adjuvant-like effects by binding to various pattern recognition receptors, such as Toll-like receptors 2 and 4 (TLR2 and TLR4) and receptors for advanced glycosylation end products (AGE) [19]. These ICD-associated DAMPs recruit APCs to the tumor site and increase their capacity to engulf, process and cross-present tumor-derived antigens to T cells. This favors the induction of tumor-specific adaptive immunity.

Tumor-infiltrating lymphocytes (TILs) and tertiary lymphoid structures (TLSs) have favorable prognostic value in TNBC [20]. The immune response in tumors is mainly dependent on adaptive immunity mediated by T cells [21]. CD8^+^ T cells evolve and kill tumor cells by releasing perforin, granzymes and INF-γ [22]. DCs are central to the initiation of primary immune responses and they are the only APCs capable of stimulating T cells, making them pivotal in the generation of adaptive immunity against TNBC [23]. TNBC patients with higher levels of TILs and TLSs in the TME have better clinical outcomes [24,25]. Additionally, recent reports suggest that CD11c+ DCs have a strong correlation with the levels of TILs and TLSs in TNBC [26]. Therefore, therapeutic modalities that facilitate activation and recruitment of CD11c+ DCs has immense value. In addition, CD11c is a type I transmembrane protein that is expressed on monocytes, granulocytes, a subset of B cells, DCs and macrophages. Highly enriched CD11c+ DCs can be differentiated from mouse bone marrow [27] to assess their activation or maturation status by various agents ex vivo.

Tumor necrosis factor-α (TNF-α) activates TNF receptors and induces either cellular proliferation, survival, differentiation or cell death, depending on the contextual presence of other signaling molecules [28]. Approximately 28% of all cancers have been shown to be susceptible to direct induction of cell death by TNF-α [29]. The antitumor activity of TNF-α is mediated by the cell surface receptor TNFR1 which activates caspase-8, culminating in apoptosis [28]. An important downstream regulator of TNFR signals is the enzyme called receptor interaction protein kinase 1 (RIPK1) [28]. The type of ubiquitination mark on RIPK1 determines whether a cell undergoes proliferation or death [30]. It is well known that TNBC can become resistant to various chemotherapeutic drugs, including anthracycline-based compounds [31,32]. Therefore, activating TNFR with TNF-α, in combination with pan caspase inhibitors, such as z-VAD-fmk, to induce cell death by necroptosis, could represent a therapeutic approach in treating TNBC [33]. Indeed, it has been reported that in MDA-MB-231 TNBC cells, when incubated with small-molecule antagonists of inhibitors of apoptosis (IAPs), also known as second mitochondria-derived activator of caspases (SMAC) mimetics, activate the autocrine production of TNF-α by activating RIPK1 [34]. In addition, several studies in different types of cancers have shown the potential of ICD inducers to increase the levels of pro-inflammatory cytokines, including TNF-α, to produce an immunogenic TME [3,35,36,37,38,39,40]. Although TNF-α is known to cause systemic inflammation [28], the selective and restricted activation of the TNF-α pathway in cancer cells, if achieved, could reprogram DCs in the TME to target cancer cells.

In the present *in vitro* study, we report that a novel small, thienopyrimidine-derived molecule, TPH104, produces a non-apoptotic cell death in TNBC cells [41]. MDA-MB-231 cells were highly sensitive to TPH104, while non-cancerous cells were relatively resistant to TPH104. Unlike most small molecules, TPH104 induced rapid membrane permeabilization and had characteristics of non-apoptotic cancer cell death in culture. In addition, TPH104-induced ATP release, increased the cell surface translocation of CRT and activated the endogenous TNF-α pathway by imparting immunogenic characteristics to dying MDA-MB-231 cells. Furthermore, the contents released by such dying tumor cells stimulated the maturation of DCs and upregulated the inflammatory gene expression. These effects of TPH104 in TNBC cells activate DCs and suggest that TPH104 may represent a novel therapeutic candidate in the treatment of aggressive breast cancers.

## 2. Materials and Methods

### 2.1. Chemicals, Antibodies and Reagents

Recombinant mouse (rm) GM-CSF (#BMS325), 2-mercaptoethanol (# 21985-023) and a high-capacity cDNA reverse transcription kit (# 4368814) were obtained from Life Technologies (Grand Island, NY, USA). FITC anti-mouse MHC class II antibody (#11-5321-81) and calreticulin antibody (# PA3-900) were obtained from Thermo Fisher Scientific, Inc. (Waltham, MA, USA). Human TNF-α neutralizing antibody (#7321S), anti-rabbit IgG (H+L) Fab2-Alexa Flour 488 (#4412), Immunofluorescence blocking buffer (#12411P), Immunofluorescence antibody dilution buffer (#12378P) and 16% formaldehyde (#12606P) were obtained from Cell Signaling Technology (Danvers, MA, USA). A human TNF-α ELISA kit (# 43204), Tru Stain fcX (anti-mouse CD16/32), Clone: 93 (#101320), PE/Cy7 anti-mouse CD86 antibody, Clone: GL-1 (#105013), and APC anti-mouse CD11c+ antibody, Clone: N418 (#117310), were obtained from BioLegend. Lipopolysaccharide (LPS) (*Escherichia coli* serotype 0111: B4) (# L4391), necrostatin-1 (# N9037), ATP (# A7699) and doxorubicin hydrochloride (# D1515) were purchased from Sigma (St. Louis, MO, USA). TPH104 was synthesized in-house, and the synthetic scheme and its characterization will be published elsewhere. 3-(4,5-dimethylthiazol-2-yl)-2,5-diphenyltetrazolium bromide (MTT) (# T-030-10) was obtained from Gold Biotechnology (St. Louis, MO, USA). Mitoxantrone (# S2485) was obtained from Selleck Chemicals (Houston, TX, USA). SYBR Green supermix (# 95054) was obtained from Quantabio (Beverly, MA, USA). Fluorosheild mounting medium with 4’,6-diamidino-2-phenylindole (DAPI, # ab104139) was obtained from Abcam (Cambridge, MA, USA). A bicinchoninic acid assay (BCA) protein quantitation kit (# 786-570) was obtained from G-Biosciences (St. Louis, MO, USA). SP600125 (#10010466), a potent and reversible inhibitor of c-Jun N-terminal kinase (JNK), was obtained from Cayman chemicals, Ann Arbor, MI, USA.

### 2.2. Cell Culture

MDA-MB-231 and BT-20 cell lines were obtained from late Dr. Gary Kruh (University of Chicago, Illinois). Human primary epithelial gingival keratinocytes (HPK), and normal colon fibroblast CRL1459 cells were purchased from American Type Culture Collection (ATCC#PCS-200-014, and ATCC#CCD-18Co—CRL-1459, respectively, Manassas, VA, USA). These cells were grown as an adherent monolayer with Dulbecco’s Modified Eagle’s Medium (Corning, Tewksbury, MA, USA) supplemented with 10% fetal bovine serum (Biofluids Technologies, Port Richey, FL, USA), 2 mM of L-glutamine and 100 U/mL of penicillin and 100 µg/mL of streptomycin (Thermo Fisher Scientific, Inc., Waltham, MA, USA), as previously described [42]. HPKs cells were cultured in dermal cell basal medium (ATCC# PCS-200-030) supplemented with factors from Keratinocyte growth kit (ATCC# PCS-200-040) as per manufacture’s protocol. These cells were cultured in a humidified incubator containing 5% CO_2_ at 37 °C. All cells were assessed and confirmed to be free of fungi and mycoplasma. Cells were obtained from frozen stocks and cell passaging (up to P4) was performed at 80% cell confluency trypsin + 2.2 mM EDTA in phosphate-buffered saline (PBS).

### 2.3. Cell Cytotoxicity Assay

The MTT assay was performed to determine the cytotoxic efficacy of TPH104, as described previously [43]. Briefly, in a 96-well flat-bottom cell culture plate, cells were seeded at a density of 3 × 10^3^ cells/well (in 100 µL) and incubated overnight. The next day, cells were incubated with TPH104 in 100 μL to achieve a final concentration of 0.05, 0.1, 0.25, 0.5, 1, 5, 10, 25 or 50 µM in a total volume of 200 μL. After 72 h of incubation, 20 µL of 4 mg/mL of MTT was added to the plate and incubated for 3 h at 37 °C. Following incubation, the supernatant was removed, and the insoluble formazan precipitates were solubilized in 150 µL of dimethyl sulfoxide (DMSO) and mixed on a plate shaker for 10 min. The absorbance was measured with a micro-plate reader (Synergy H1, Biotek, VT, USA) at 560 nm. IC_50_ values were calculated from the nonlinear regression curve fit of log of inhibitor versus the dose response parameter in the GraphPad Prism (Version 8.4.3, GraphPad Software, San Diego, CA, USA) program.

### 2.4. IncuCyte™ Live-Cell Morphology Study

The morphological changes induced by TPH104 on MDA-MB-231 cells were visualized through the Incucyte^®^ S3 Live-Cell Analysis System (Ann Arbor, MI, USA). Briefly, MDA-MB-231 cells were seeded at 4000 cells/well in a 96-well plate and incubated overnight in a 37 °C, 5% CO_2_ incubator. Then, the cells were treated with varying concentrations (0, 0.5, 1 and 5 µM) of TPH104 and placed in the Incucyte system. The images were then captured at different time points using the integrated IncuCyte S3 software version 2020B.

### 2.5. Preparation of the Conditioned Media (CM) from Death-Primed MDA-MB-231 Cells

The MDA-MB-231 cells were cultured in 60 mm culture dishes in the presence of vehicle (<0.1% DMSO), TPH104 (1 or 5 µM), and MX (1 µM). The supernatants were collected after 0, 24 and 48 h of incubation and ATP levels in the supernatants were measured using CellTiter-Glo^®^ kits (Promega, Madison, WI, USA). The supernatants collected were concentrated by centrifugation in 10,000 MWCO (molecular weight cutoff) centrifuge tubes (Millipore Sigma, Burlington, MA, USA) at 3000× *g* for 30 min at 4 °C and stored at −80 °C. VC-CM represents the vehicle control (VC) conditioned media (CM) of MDA-MB-231 cells incubated with vehicle, whereas TPH104-1-CM describes the CM of MDA-MB-231 cells incubated with TPH104 (1 µM); TPH104-5-CM, the CM of MDA-MB-231 cells incubated with TPH104 (5 µM); and MX-1-CM, the CM of MDA-MB-231 cells incubated with MX (1 µM). A schematic representation of the protocol is provided in Appendix A.

### 2.6. The Generation of Mouse Bone Marrow-Derived Dendritic Cells (BMDCs)

Mouse bone marrow-derived dendritic cells (BMDCs) were generated as previously described [44], with minor modifications. Briefly, mouse BM cells were removed from the marrow of femurs from sacrificed C57BL/6 mice and depleted of erythrocytes by hypotonic lysis using 0.8% ammonium chloride. BM cells were cultured in RPMI-1640 (Life Technologies, Grand Island, NY, USA), supplemented with 5% heat-inactivated fetal bovine serum (# BT-101-500-D; Biofluid Technologies LLC, Port Richey, FL, USA), L-glutamine (0.03%), sodium pyruvate (0.4 mM), 2-mercapthoethanol (50 mM), mGM-CSF (20 ng/mL) and penicillin/streptomycin in 60 mm culture dishes (2.5 × 10^6^ cells/dish). Fresh culture medium was replaced on days 3 and 5. On day 6, loosely held, non-adherent cells were harvested and reseeded at a density of 1.5 × 10^6^ cells per well in a 6-well plate, incubated with conditioned media harvested from death-primed MDA-MB-231 cells, in a ratio of 1:7.5 and cultured for 24 h before harvest, were used for flow cytometry or PCR assays. LPS was used as a positive control for stimulation of DCs.

The BM cells were generated from the femurs of male C57BL/6 age-matched mice, purchased from Taconic Biosciences, Inc. (Hudson, NY, USA) and maintained on an AIN-76-based semi-purified diet from Harlan-Teklad (Madison, WI, USA). The animal protocols in this study were approved by the Institutional Animal Care and Use Committee and the Institutional Biosafety Committee at the University of Toledo.

### 2.7. Flow Cytometry

Cells were counted after trypan blue staining using a hemocytometer. Approximately 5 × 10^5^ cells were suspended in 100 µL of flow buffer (phosphate-buffered saline (GE Life Sciences), containing 100 IU/mL of penicillin (Corning), 100 µg/mL of streptomycin (Corning) and 2% (*v*/*v*) fetal bovine serum (Biofluid Technologies). The staining of the extracellular cell surface antigens was performed by incubating cells with 0.25 µg of Fc block (Tru Stain fcX; BioLegend) for 10 min, followed by 30 min of incubation at 4 °C, with the following antibodies: 0.125 µg of FITC-conjugated MHC-II (Invitrogen), 1 µg of PE-Cy7-conjugated CD86 (BioLegend) and 0.25 µg of APC-conjugated CD11c+ (BioLegend). Subsequently, the cells were washed with flow buffer and collected on a BD Accuri C6 plus Flow Cytometer (BD Biosciences), including unstained and single-stained controls. Single cells were obtained by gating on FSC-A versus FSC-H, followed by SSC-A versus APC-A. APC-CD11c+ gated cells were further resolved into distinct Hi and Lo populations of CD86 and MHC-II. The data analysis was performed using FlowJo (Ashland, OR, USA) software.

### 2.8. Extracellular ATP Release Measurements 

Cells were incubated at 37 °C with Opti-MEM (Life Technology) containing vehicle or the test compounds. Supernatants were collected at 24 and 48 h and ATP levels were measured using the CellTiter-Glo ATP assay system kit (Promega, Madison, WI, USA). ATP levels were calculated based on an ATP standard curve.

### 2.9. Measurement of TNF-α

The MDA-MB-231 cells were seeded at a density of 6.0 × 10^5^ cells in 6 cm culture dishes (Falcon). The cells were cultured overnight and the following day, culture media were replaced with fresh media containing either vehicle (<0.1% DMSO) or TPH104, at a final concentration of 1 or 5 µM, for 48 h and the supernatant and cells were collected and stored at −80 °C. TNF-α levels were measured using an ELISA kit (BioLegend, San Diego, CA, USA) according to the manufacturer’s instructions, and the values were normalized to the total cellular protein content.

### 2.10. RNA Extraction and Real-Time Quantitative PCR 

Cells were harvested in 1 mL Tri Reagent (Thermo Fisher Scientific) after incubation with vehicle or the test compounds, following gentle washing with ice-cold PBS, and stored at −80 °C until RNA extraction was performed. RNA extraction was conducted according to the manufacturer’s protocol, with modifications, as previously described [45]. Subsequently, RNA was reverse transcribed to cDNA using a High-Capacity cDNA Reverse Transcription Kit (Life Technologies). cDNA from various cell samples was amplified with specific primers (human *TNF-α*: 5′- GCGTTCTTTGCTGGCTGCACAA -3′ and 5′- CTCCAAGCAATGCCTGTAGTCTC -3′; mouse *TNF-α*: 5′- ACTCCAGGCGGTGCCTATGT -3′ and 5- AGTGTGAGGGTCTGGGCCAT -3′; mouse *IL1β*: 5′ - TTGACGGACCCCAAAAGATG -3′ and 5′- AGAAGGTGCTCATGTCCTCAT3′; mouse *IL-6*: 5′- ACAACGATGATGCACTT-3′ and 5′- CTTGGTCCTTAGCCACT -3′; mouse *IL-12A(p35)*: 5′- CCTCAGTTTGGCCAGGGTC -3′ and 5′- CAGGTTTCGGGACTGGCTAAG -3′; mouse *IL-12B(p40)*: 5′- GGAAGCACGGCAGCAGAATA -3′ and 5′- AACTTGAGGGAGAAGTAGGAATGG -3′) and the data were normalized to mouse *β-actin*: 5′- CCTCTATGCCAACACAGTGC -3′ and 5′- ATACTCCTGCTTGCTGATCC -3′, and human *β-actin*: 5′- CCTAAAAGCCACCCCACTTCTC -3′ and 5′- ATGCTATCACCTCCCCTGTGTG -3′, in a CFX Connect Real-Time PCR Detection System (Bio-Rad, Hercules, CA, USA). Transcript abundance was analyzed, and 2^−ΔΔCt^ was calculated to represent changes in gene expression relative to controls [46].

### 2.11. Immunofluorescence for Calreticulin

The MDA-MB-231 cells were seeded on 12.0 mm round sterile glass coverslips (1.5 × 10^5^ cells/well) in a 12-well plate. Cells were cultured overnight and incubated with either vehicle, TPH104 (5 µM for 6 h) or mitoxantrone (1 µM for 2 h). After incubation, cells were placed on ice, washed twice with PBS and fixed with 4.0 % PFA in PBS for 5 min. Cells were then washed twice in PBS and blocked for 1 h with blocking buffer (Cell signaling). Subsequently, a calreticulin polyclonal antibody (Invitrogen), diluted in antibody dilution buffer (1:100), was added for 1 h at room temperature. After washing three times in ice-cold PBS, the cells were incubated for 1 h with goat anti-rabbit IgG (Fab2) Alexa Flour 488 (Cell Signaling), diluted 1:1000 in cold antibody dilution buffer. The cells were washed three times with PBS, dried and mounted on slides with the mounting medium, including DAPI (Abcam), and allowed to cure overnight. Fluorescence microscopic assessment was performed using a Zeiss Axiovert 40 CFL inverted fluorescence microscope (Carl Zeiss, Germany). Fluorescence intensity was quantified using NIH image J software.

### 2.12. Statistical Analysis 

Experiments were repeated at least three times or as indicated in the legends. Data are expressed as the mean ± SEM. Either an unpaired one-tailed *t*-test or unpaired two-tailed *t*-test was used and the results were analyzed using GraphPad Prism (GraphPad Software). The a priori significance level was *p* ≥ 0.05.

## 3. Results

### 3.1. Cytotoxicity of TPH104 Is Selective to MDA-MB-231 Cells 

In order to determine the IC_50_ value of TPH104 for MBA-MB-231 and BT-20 cells, were incubated with various concentrations of TPH104 for 72 h and their viability was determined using the MTT assay. The IC_50_ value of TPH104 (Figure 1A), on TNBC cells for MDA-MB-231 was 140 nM and BT-20 cells were 270 nm, respectively, while it was higher in the normal gingival epithelial HPKs (i.e., 15.23 μM) and normal colon fibroblast CRL-1459 (i.e., 24.7 μM (Figure 1B). This suggest that TPH104 has > 100-fold selectivity for normal cells compared to TNBC MDA-MB-231 cells.

Incubation with TPH104 (0.5, 1 and 5 µM) produced a concentration-dependent decrease in cell density (Figure 1C,D). TPH104 induced cellular swelling and an increase in the permeability of the MDA-MB-231 cell membrane (Figure 1E). In addition, the incubation of MDA-MB-231 cells with 10 µM of TPH104 produced large, stable and continuously expanding membrane blebs and swelling (Figure 1E) (Appendix A) and plasma membrane permeabilization, indicative of necrotic cell death [47]. The occurrence of membrane hyperpermeability indicates decreased cell viability. Overall, our results indicate that TPH104 induced selective non-apoptotic type of cell death in MDA-MB-231 and BT-20 cells.

### 3.2. TPH104-Mediated Death Primes MDA-MB-231 Cells to Express Biomarkers of Immunogenicity

Chemotherapeutic compounds have been shown to induce ICD [9]. Important biomarkers of ICD include cell surface exposure of calreticulin by death-primed cells [7] and the release of ATP and HMGB1 as they undergo death [9]. To assess the effect of TPH104 on the immunogenic modulation of MDA-MB-231 cells, we measured ATP levels in the culture supernatant of the cells. MDA-MB-231 cells incubated with 5 µM of TPH104 for 24 or 48 h released a significantly higher amount of ATP into the culture supernatant compared to the vehicle control (Figure 2A). Furthermore, the expression of CRT was significantly increased in MDA-MB-231 cells incubated with TPH104 (5 µM) and mitoxantrone (MX—1 µM, positive control) compared to cells incubated with the vehicle (Figure 2B). At higher magnifications, the concentrated fluorescence of CRT in the plasma membrane microdomains was more prominent following the incubation of MDA-MB-231 cells with either TPH104 or MX compared to the control cells incubated with the vehicle (Appendix A). Our in vitro results suggest that TPH104 can increase the expression of immunogenic biomarkers in death-primed MDA-MB-231 cells.

### 3.3. TPHI04 Induces the Production of TNF-α by MDA-MB-231 Cells 

MDA-MB-231 cells can produce TNF-α after the inhibition of cytoplasmic inhibitors of apoptosis in the RIP1 kinase-dependent pathway [34]. Given that our morphological studies indicated onco-necrosis, this led us to hypothesize that TPH104 may increase the levels of TNF-α in MDA-MB-231 cells. TPH104 (1 or 5 µM) significantly increased (*p* > 0.01) the levels of TNF-α mRNA, as determined by qPCR in MDA-MB-231 cells (Figure 3A). Compared to the control, the cells incubated with 1 µM of TPH104 for 24, 36 or 48 h produced a 48, 81 and 62% increase in TNF-α mRNA expression, respectively. This trend was also observed at higher concentrations of TPH104 (5 µM) which produced a 62, 112 and 108% increase in TNF-α mRNA levels at 24, 36 or 48 h, respectively, as compared to the control (Figure 3A). Furthermore, TPH104 (1 and 5 µM) significantly increased (*p* ≥ 0.0001) the level of the TNF-α protein to 16.7 and 77.1 pg/mg, respectively (basal levels of TNF-α were 0.26 pg/mg of total protein in MDA-MB-231 cells incubated with the vehicle) (Figure 3B). As the data suggest that TPH104 can activate the TNF-α pathway in MDA-MB-231 cells, we blocked TNF-α signaling with a TNF-α-neutralizing antibody (0.2, 1 and 5 µg/mL) in MDA-MB-231 cells incubated with 1, 5 or 10 µM of TPH104. There was no significant decrease in the TPH104-induced cytotoxic effects (Figure 3C). These results indicate that TPH104 activated the TNF-α pathway and TNF-α does not mediate cell death in MDA-MB-231 cells.

### 3.4. TPH104 Elicits the Release of Factors from MDA-MB-231 Cells That Induce the Maturation of DCs

Dendritic cells respond to external stimulatory signals (PAMPs and DAMPS) and upregulate the surface expression of maturation markers such as MHC-II, CD86 and CD80 [48]. Such an activation of DCs by stimulatory signals including TNF-α increases CD8^+^ T cell-mediated antitumor immunity that is essential for the efficacy of cancer immunotherapy [49]. Since TPH104 induced the production of immunogenic biomarkers including TNF-α in MDA-MB-231 cells, it prompted us to investigate the effect of supernatants collected from MDA-MB-231 cells that were death primed by TPH104. Therefore, BMDCs were incubated ex vivo with a 7.5:1 ratio of BMDC media and conditioned media collected from either the vehicle (VC-CM)-, 1 µM TPH104 (TPH104-1CM)-, 5 µM TPH104 (TPH104-5-CM)- or 1 µM mitoxantrone (MX-1-CM)-treated MDA-MB-231 cultures for 48 h and analyzed by flow cytometry (Figure 4). The percentage of the CD11c+ population did not vary significantly among VC-CM-, TPH104-1-CM- and TPH104-5-CM-treated BMDCs (Figure 4B). However, their activation profile was altered as indicated by the levels of MHC-II and CD86 (Figure 4C,D). A marked increase is evident for levels of both MHCII and CD86 in BMDCs treated with TPH-1-CM and TPH-5-CM in comparison to VC-CM-treated BMDCs. We observed different activation states of CD11c+ BMDCs such as MHC-II^Hi^, MHC-II^Lo^, CD86^Hi^ and CD86^Lo^ populations. Upon incubation with TPH104-1-CM or TPH104-5-CM, MHC-II^Hi^ and CD86^Hi^ populations were significantly increased with a corresponding decrease in MHC-II^Lo^ and CD86^Lo^ populations in CD11c+ BMDCs in comparison to the VC-CM-treated counterpart. Further, incubation of BMDCs with a TLR-4-specific ligand—LPS—shifted the MHC-II^Lo^ and CD86^Lo^ populations completely (>90%) towards MHC-II^Hi^ and CD86^Hi^ (Figure 4E), which was used a quality control experiment to evaluate the maximal activation of DCs. Furthermore, MX-1-CM that was collected from the mitoxantrone (1 µM)-treated MDA-MB-231 culture used as a positive control for immunogenic cell death induction significantly increased the expression of CD11c+ (*p* < 0.01) compared to BMDCs incubated with VC-CM. In addition, BMDCs treated with MX-1-CM significantly increased MHC-II^Hi^ (*p* < 0.05) and CD86^Hi^ (*p* < 0.05) populations in comparison to VC-CM. All these results assert that TPH104-treated (1 and 5 µM) MDA-MB-231 cells release factors that can induce expression of maturation markers such as MHC-II and CD86 DCs.

### 3.5. MDA-MB-231 Cells Stressed by TPH104 Release Factors to Modulate BMDC Cytokine Expression

The conditioned media, TPH104-1-CM, TPH104-5-CM and MX-1-CM, prepared from MDA-MB-231 cells differentially modulated the cytokine expression of BMDCs. TPH104-5-CM significantly increased the expression of *TNF-α* (*p* < 0.01) (Figure 5A), *IL-6* (*p* < 0.05) (Figure 5B) and *IL-12A* (*p* < 0.05) (Figure 5D) by 67, 327 and 45%, respectively, compared to VC-CM-incubated BMDCs. The positive control, MX-1-CM, produced a non-significant increase in the mRNA levels of *TNF-α* (Figure 5A) and *IL-6* (Figure 5B) but significantly increased the levels of *IL-12A* (Figure 5D) (*p* < 0.01) and *IL-12B* (*p* < 0.01) (Figure 5E) by 105 and 166%, compared to cells incubated with the vehicle. The levels of *IL-1β* did not change significantly in BMDCs incubated with different CM. These data further indicate that TPH104 and MX-primed death of MDA-MB-231 cells upregulated a subset of cytokines relevant to tumor immunity.

In Figure 6, we summarize our findings for TPH104-mediated cell death in MDA-MB-231 cells. TPH104 induced a unique, non-apoptotic cell death in MDA-MB-231 cells and increased the expression of immunogenic biomarkers such as ATP and calreticulin. Furthermore, TPH104-mediated death significantly increased TNF-α levels. The TPH104-mediated changes in TNBC can be immunogenic and the factors released by dying cells induce DC maturation and an inflammatory gene expression.

## 4. Discussion

The selective targeting of cancer cells, by release of inflammatory cytokines and upregulation of biomarkers of ICD by dying cancer cells, can increase the killing of cancer cells and decrease tumor cell proliferation by immune cells in the TME [50]. In the present study, the incubation of MDA-MB-231 cells with TPH104 induced the autocrine production of TNF-α from dying MDA-MB-231 cells and increased the expression of the immunogenic markers ATP and CRT. 

Cancer cells undergoing ICD mobilize intracellular CRT to the outer leaflet of the plasma membrane to facilitate tumor cell recognition and phagocytosis by DCs and subsequent T cell-mediated elimination of cancer cells [7]. The results of our study indicate that TPH104 induced the translocation of CRT to the plasma membrane of MDA-MB-231 cells *in vitro* (Figure 2B and Appendix A). The upregulation of CRT by MX has been previously reported, and our results are consistent with those of a previous study [51]. CRT is an endoplasmic reticulum (ER)-resident calcium binding protein that plays a key role in Ca^2+^ homeostasis [52], regulation of nuclear protein transport [53], nuclear steroid receptor signaling [54] and integrin signaling [55] and is a molecular chaperone for the correct folding of polypeptides retained in the ER [54]. The exact mechanism for the translocation of CRT remains to be determined, although published studies indicate that the chemotherapy-mediated ER stress response induces the phosphorylation of eIF2α to facilitate the exocytosis of CRT [17]. Therefore, it is possible that therapies that induce eIF2α phosphorylation, such as inhibitors of the GADD34/PP1 complex, which normally dephosphorylates peIF2α [7], could be useful in the induction of ICD. In addition to CRT externalization, TPH104 (5 µM) significantly increased the levels of extracellular ATP in the culture supernatant of MDA-MB-231 cells (Figure 2A). The active release of ATP by dying cancer cells can sensitize DCs by activating purinergic receptors (P2RX7 and P2RY2) [56]. 

The induction of apoptosis, [7], γ-irradiation cytotoxicity [7,57] and necroptosis (TNF-α plus Z-VAD-fmk and SMAC mimetics) mobilize plasma membrane CRT and release ATP and HMGB1 [58], producing ICD in cancer cells. Although the detailed modality of cell death induced by TPH104 is described elsewhere (manuscript in submission), the cellular morphological features indicate that it induces non-apoptotic cell death characterized by cell swelling, non-significant nuclear condensation and CRT externalization and ATP release. Due to technical limitations, we could not measure HMGB1 levels. Therefore, it is not clear if TPH104 elicits the release of HMGB1 or other alarmins. Furthermore, different cell death programs produce distinct reshuffling processes of the plasma membrane and release intracellular content that plays a crucial role in dying/dead cell-induced immunogenicity [59]. 

The precise molecular mechanisms that induce secretion of ATP during ICD need to be explored further. As per the current understanding, the apoptotic machinery (caspase3-dependent cleavage of pannexin1 channels for ATP efflux during ICD) [60] and the autophagy competency of cancer [56] are necessary for active secretion of ATP during ICD. This might contradict our findings of ATP release during a non-apoptotic cell death induced by TPH104 in an autophagy incompetent TNBC [61]. However, a caspase/pannexin-1-independent mechanism of ATP release is possible [62]. Cell death induced by TPH104 resembles anoikis—a type of apoptosis induced by a lack of correct cell/ECM attachment that depends on caspase activation and mitochondrial permeabilization [63,64]. Unpublished data from our lab indicate no activation of caspases or annexin staining in TNBC cells treated with TPH104. In view of this, it is unlikely that cells are undergoing anoikis. In addition, during apoptotic cell death, protein synthesis is actively abolished through caspase-mediated destruction of the translational machinery of cells. Contrastingly in TPH104-induced cell death, corpses of MDA-MB-231 still produce TNF-α, indicating a non-apoptotic cell death induced by TPH104. A similar phenomenon was reported recently where necroptotic corpses of cells can continue to produce protein via an intact ER network even after loss of plasma membrane integrity [65]. The mechanism of cell death induced by TPH104 in MDA-MB-231 is extraordinarily complex and factors driving the ATP release during such a process need a more thorough understanding.

A tumor is a heterogeneous mass of cancer cells, with stromal features such as vasculature, fibroblasts and infiltrating immune cells, which collectively form the TME [66]. The TME is highly variable among tumors and is vital for tumor growth, migration and evasion of the immune response [67]. Furthermore, the TME maintains immunosuppressive conditions by (1) upregulating the expression of immune checkpoint-related molecules, such as the anti-programmed cell death protein 1/programmed cell death ligand 1 (PD-1/PD-L1) [68], the anti-cytotoxic T lymphocyte-associated antigen-4 (CTLA-4), T cell immunoglobulin and mucin-domain containing-3 (TIM-3) and lymphocyte activation gene-3 (LAG-3) [69]; (2) increasing the population of immuno-regulatory cells; (3) the hypoxic nature of tumors and the presence of hypoxia-driven metabolites, such as lactate, in the TME [70]; and (4) decreasing the efficacy of effector cells by a variety of mechanisms, including inhibition of DC maturation [71]. Therefore, targeting the immunosuppressive TME with ICD inducers can increase adaptive immune priming and improve the efficacy of effector cell killing and cytotoxic therapies. 

TPH104-mediated death of MDA-MB-231 cells significantly increased the expression of *TNF-α* mRNA (Figure 3A) and the release of TNF-α (Figure 3B) into the cell culture supernatant. Necroptotic cells can continue to produce cytokines, such as TNF-α, to further optimize necroptotic cell death, thereby increasing their immunogenicity [65,72]. In addition, recent reports indicate that activation of the necroptotic pathway is associated with inflammatory transcription and cell death simultaneously [72,73]. Furthermore, during certain forms of cell death, even after cells are permeabilized, they can continue to synthesize proteins such as TNF-α [65]. Our results are consistent with a previous study indicating that MDA-MB-231 cells produce TNF-α in response to certain chemotherapeutic drugs [34]. In our study, TPH104 increased the expression of *TNF-α* mRNA, particularly in transformed cells, which could increase the immune response to the tumor in the TME. Our results indicate that DCs exposed to CM from MDA-MB-231 cells incubated with TPH104 (TPH104-CM) induced the maturation of BMDCs (Figure 4). The maturation of DCs can be attributed to TNF-α (Figure 3A,B) and other DAMPs in the CM. Earlier studies have shown that the presence of TNF-α during and after ICD increased MHC II expression on APCs and facilitated T cell differentiation and NK cell activation [50,74]. Furthermore, when ex vivo BMDCs were incubated with TNF-α induced their maturation, based on the increased expression of maturation markers, including MHC-II and CD86 [75]. Similarly, we observed an increased expression of MHC-II and CD86 (Figure 4) in DCs produced by TPH104-CM that could, in part, explain the increase in TNF-α levels in the CM. Furthermore, TPH104 (5 µM) death-primed MDA-MB-231 cells, such as MX-incubated cells, release biomolecules that increase inflammatory gene expression, notably, TNF-α and IL-6, which are downstream of TLRs in DCs. 

DCs express both MHC I and II that play a role in the activation of CD8 and CD4 T cells, respectively [11]. DCs phagocytose antigens in the cytosol for cross-presentation of tumor antigens to MHC I, a prerequisite to prime cytotoxic CD8^+^ T cells [76]. The increase in specific types of mature DCs in the TME has been correlated with favorable outcomes for the stimulation of cytotoxic T cells [77]. It is important to note that multiple subsets of DCs have different effects on tumor immunology [66,78]. Furthermore, antitumor adaptive immune responses are predicted based on the processing of tumor antigens by APCs such as DCs, which, when subjected to appropriate activation signals, migrate to secondary lymphoid tissue, where they prime naïve T cells [76] for targeting tumors bearing those antigens. Therefore, the activation of DCs by extracts of dying MDA-MB-231 cells is a key feature of TPH104-mediated TNBC cytotoxicity that could induce adaptive immune priming against the tumor. 

In addition to presenting antigens and co-stimulatory signals being required for the activation of T cells, DCs induce cells to differentiate to IFN-γ-producing (Th1 phenotype) or IL-4-producing cells (Th2) [79]. Th1 differentiation is driven by the presence of IL-12 in the microenvironment, where antigen presentation occurs [80,81]. DCs are the major source of IL-12, which is an important stimulatory factor for NK cell activation [82]. Our results indicate that DCs incubated with TPH104-5-CM and MX-1-CM significantly increased the expression of IL-12A and IL-12B, respectively (Figure 5). This is a novel and important finding due the fact that IL-12 is a multifunctional cytokine that induces IFN-γ γ production and stimulates the growth of both T and NK cells, increasing Th1-type helper T cell responses and inhibiting neovascularization [83]. Furthermore, the release of IL-12 by DCs suppresses tumor neovascularization [84]. The efficacy of CM to induce the expression of IL-12 by DCs could be beneficial in facilitating T cell-mediated immunity in the TME and inhibit tumor angiogenesis, a significant feeder of tumor growth and metastasis [85]. 

The release of inflammatory mediators is important for the synergy between non-apoptotic cell death and immune therapy [2]. Consistent with this concept, our results indicate that TPH104 not only killed MDA-MB-231 cells via a non-apoptotic pathway but also caused ICD, accompanied by the release of the pro-inflammatory mediator TNFα, to increase the adaptive immune response against cancer cells. In addition, the activation of inflammatory gene expression in DCs (Figure 5) can enrich the TME with immune-activating cytokines necessary for efficacious adaptive T cell transfer immunotherapy [49]. Furthermore, the induction of non-apoptotic cell death in TNBCs could be beneficial in overcoming chemotherapy-induced resistance. However, it must be noted that the current presentation of findings has limitations pertaining to the characterization of biomarkers of ICD generated by TNBC in response to TPH104 treatment. The mechanism of TPH104-mediated TNF-α production in TNBC cells needs to be investigated in detail. Most importantly, the *in vivo* efficacy of TPH104-mediated ICD needs to be evaluated. Overall, our results, provided they can be extrapolated to humans, suggest that TPH104 could be a potential candidate for immune adjuvant therapy.

## 5. Conclusions

In conclusion, TPH104-mediated cell death in TNBC cells showed immunogenic potential that was characterized by the release of ATP and CTR externalization. The efficacy of TPH104 to induce the sustained production of TNF-α specifically by TNBC cells could be used as an adjuvant in cancer chemotherapy as this would decrease the magnitude of immunosuppression in the TME. The non-apoptotic and immunogenic nature of TNBC cell death produced by TPH104 could be a potential treatment for aggressive TNBC cancers.

## Figures and Tables

**Figure 1 cancers-13-01954-f001:**
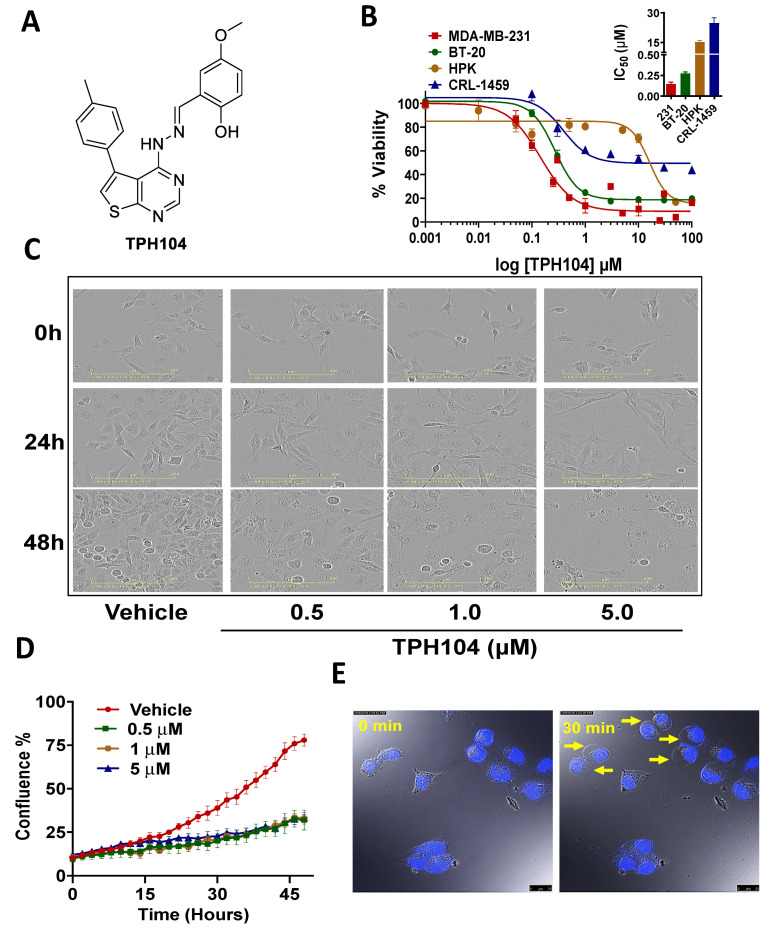
TPH104 induces non-apoptotic cell death pathway in MDA-MB-231 cells. (**A**) Chemical structure of TPH104. (**B**) Dose–response curve of TPH104 in MDA-MB-231, BT-20, HPK, CRL-1459 cells after 72 h treatment using MTT assay. In the inset, a bar graph depicting the IC_50_ ± SEM values of TPH104 for MDA-MB-231, BT-20, HPK and CRL1459 cells are shown. Data shown are mean ± S.E.M, *n* = 3 independent experiments. (**C**) Representative images showing morphological changes in MDA-MB-231 cells (20×) after incubation with either vehicle (media with ˂0.1% DMSO) or TPH104 (0.5, 1.0 and 5.0 µM) at 0, 24 and 48 h post-treatment. Scale bars: 200 µm. (**D**) Cell confluence (%) calculated using IncuCyte S3 software based on phase-contrast images of MDA-MB-231 cells from 0 to 48 h post-treatment with vehicle (media with ˂0.1% DMSO) and/ or different concentrations of TPH104. (**E**) Time-lapse microscopy reveals swelling of the plasma membrane during cell death, measured in MDA-MB-231 cells with nuclei stained with PI induced by 10 µM of TPH104. The images were derived from Appendix A.

**Figure 2 cancers-13-01954-f002:**
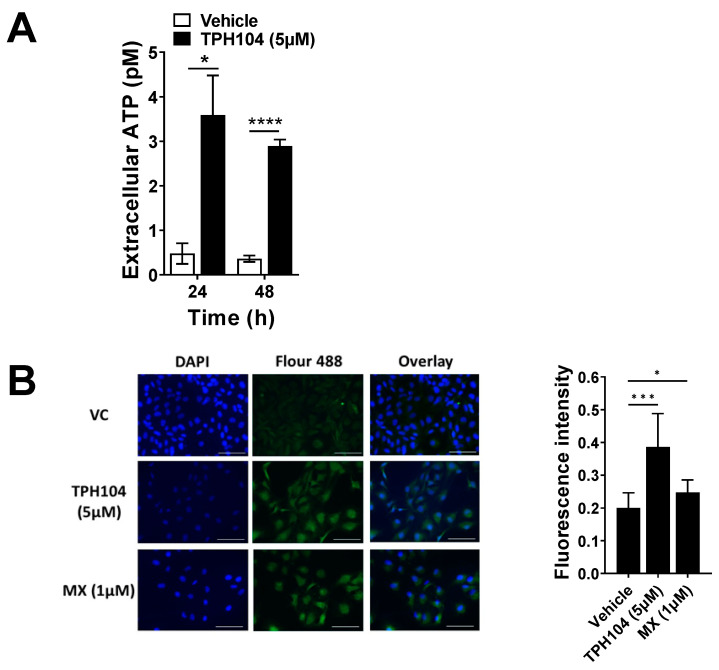
TPH104 upregulates biomarkers of immunogenic cell death in MDA-MB-231 cells. (**A**) The amount of ATP released into the extracellular medium after treatment with TPH104 for the indicated time was measured as described in Methods. *, *p* < 0.05; ****, *p* < 0.0001 compared to ATP levels in the extracellular space of MDA-MB-231 cells incubated with vehicle (<0.1% DMSO), Student’s *t* test. *n* = 4 independent experiments. (**B**) Immunofluorescence detection of cell surface CRT (pseudo-colored in green, FITC) in MDA-MB-231 cells after treatment with either vehicle (media with ˂0.1% DMSO), TPH104 (5 µM; 6 h) or MX (1 µM; 2 h). The nuclei were visualized with DAPI (40×). A representative bar graph depicting the fluorescence intensity of Vehicle, TPH104 and MX is also shown. Data represent means ± SEM. *, *p* < 0.05, *** *p* < 0.001 by Student’s *t* test.

**Figure 3 cancers-13-01954-f003:**
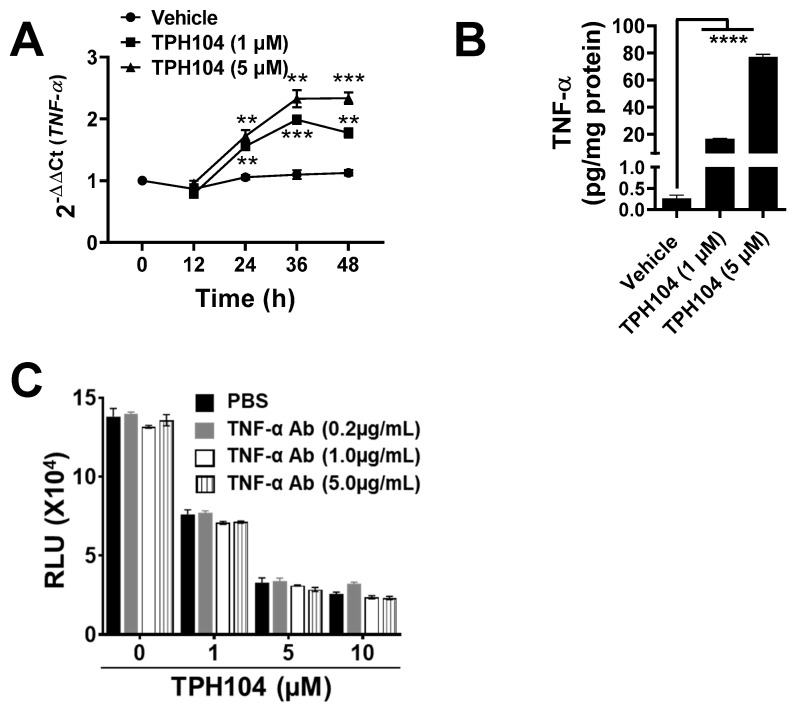
TPH104 activated endogenous production of TNF-α in dying MDA_MB-231 cells. (**A**) Real-time PCR of RNA isolated from MDA-MB-231 cells at the indicated time after treatment with either TPH104 or vehicle (media with ˂0.1% DMSO). Relative levels of the TNF-α transcript were determined compared with β-actin and the fold change was calculated by comparing with vehicle-treated cells. **, *p* < 0.01; ***, *p* < 0.001 compared to vehicle-treated cells, Student’s *t* test. *n* = 3 independent experiments. (**B**) TNF-α levels determined by TNF-α ELISA and normalized to total protein in lysate from MDA-MD-231 cells treated with either vehicle or TPH104 (1 and 5 µM) for 48 h. Data shown are mean ± S.E.M, ****, *p* < 0.0001 compared to vehicle-treated cells, Student’s *t* test. *n* = 3 independent experiments. (**C**) ATP assay of total lysates of MDA-MB-231 cells treated with TPH104 (0, 1, 5 and 10 µM) ± TNF-α-neutralizing antibody (0.2, 1, 5 µg/mL) after 48 h (the experiment was performed in technical triplicates).

**Figure 4 cancers-13-01954-f004:**
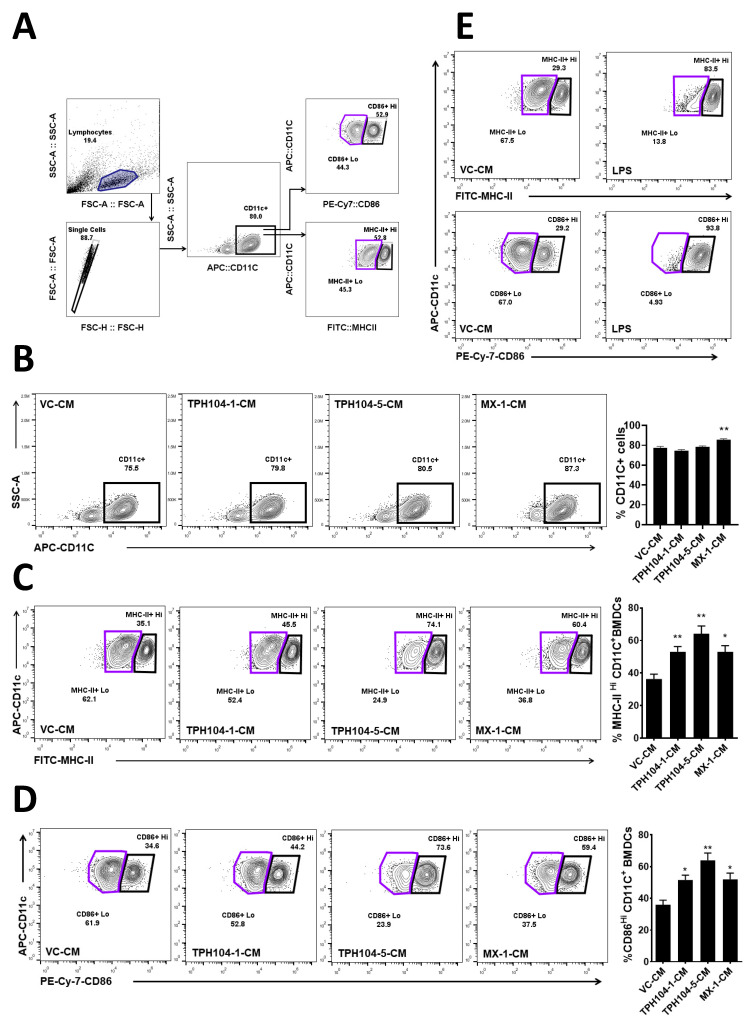
Functional maturation of mouse bone marrow-derived dendritic cells (BMDCs) induced by the conditioned media of TPH104-treated MDA-MB-231 cells. (**A**) Gating strategy used to identify MHC-II Lo/Hi and CD86Lo/Hi populations in CD11c+, CD11c+ subset of BMDCs. (**B**) Representative contour plots with their quantitation in the bar graph showing the percentage expression of CD11c+. (**C**) percentage of MHC-II Lo/Hi population and (**D**) percentage of CD86Lo/Hi population, in BMDCs incubated for 24 h in culture media spiked with conditioned media (CM) in the ratio of 7.5:1. The conditioned media were collected from cultures of MDA-MB-231 cells after 48-h treatment with either vehicle (VC-CM), TPH104; 1 µM (TPH104-1-CM), TPH104; 5 µM (TPH104-5-CM) and/or mitoxantrone; 1 µM (MX-1-CM). (**E**) MHC-IILo/Hi (TOP) and CD86Lo/Hi (BOTTOM) populations in CD11c+ subset of BMDCs cultured for 24 h with either VC-CM or LPS (100 ng/mL). Data shown in bar graph are mean ± SEM of three or more independent experiments. *, *p* < 0.05; **, *p* < 0.01 compared to percentage in DCs treated with VC-CM, Student’s *t* test.

**Figure 5 cancers-13-01954-f005:**
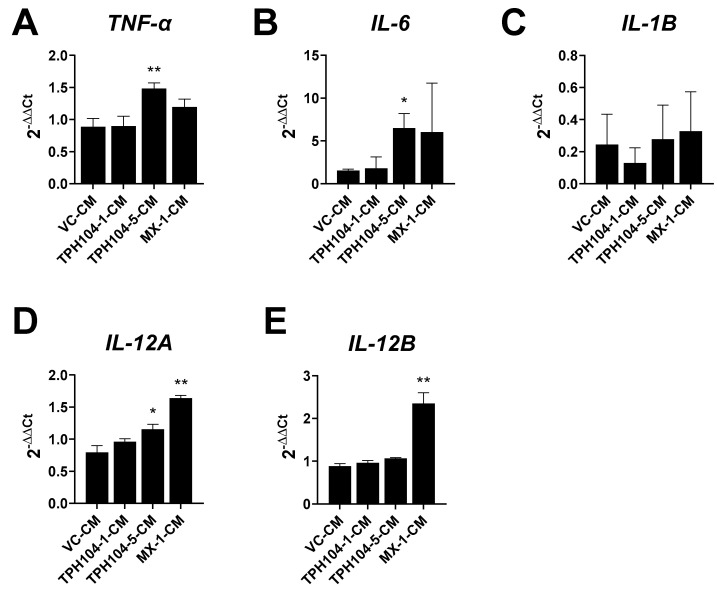
Effect of culture supernatants from death-primed MDA-MB-231 cells modulate expression of key cytokines in BMDCs. BMDCs were incubated with the indicated CM for 24 h and processed for mRNA expression by qPCR. (**A**–**E**) Average fold changes of mRNA levels of *TNF-α*, *IL-6*, *IL-1β, IL-12A* and *IL-12B* were normalized to *β-actin* mRNA level. *, *p* < 0.05; **, *p* < 0.01; compared to expression in DCs treated with CM from vehicle-treated MDA-MB-231 cells, Student’s *t* test. *n* = 4–8 independent experiments.

**Figure 6 cancers-13-01954-f006:**
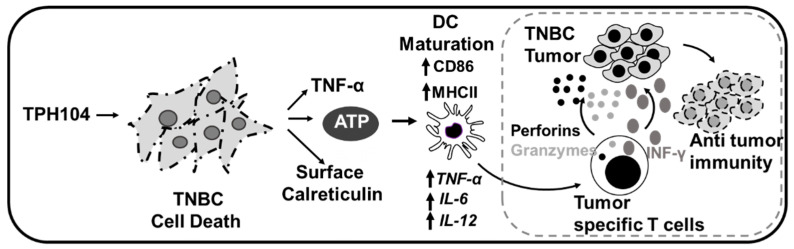
Schematic representation of TPH104-mediated immunogenic cell death in MDA-MB-231 cells. TPH104 induces a unique, non-apoptotic cell death in TNBC cells and upregulates immunogenic biomarkers such as ATP and calreticulin. Further, TPH104-mediated death prompts these cells to produce TNF-α. The cumulative effect of these manifestations culminates in maturation and activation of inflammatory genes in BMDCs. Activation of DCs can cross-prime cytotoxic T cells to affect antitumor immunity targeting TNBC tumors for destruction.

## Data Availability

Any raw data presented in this study is available on request to the corresponding author. Supporting information is available in the attached Appendix A.

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
