# Peer review of "A Novel Thienopyrimidine Analog, TPH104, Mediates Immunogenic Cell Death in Triple-Negative Breast Cancer Cells"

_cancers, 2021, doi:10.3390/cancers13081954_

Round 1
Reviewer 1 Report
Authors discovered a novel compound, thieno-pyrimidine-derived molecule TPH104, which activates cell death in the Triple negative breast cancer cells (TNBC) via complex effectors including dendritic cells (DC) and pro-inflammatory cytokine production. The paper is well written and interesting. Although several cell types were used to address the research question, only one TNBC cell line was actually used to prove the effect of TPH104 ( MDA- MB -231--- this is a serious limitation of this study)
There are several issues to address.
- “TPH104 (5 µM)” – the dose is quite high and might be non-specific. Authors should indicate what vehicle was used and discuss a possible Vehicle and TPH interactions.
- It is contradictory that TPH104 stimulated both non-apoptotic cell death and increased ATP levels from cancer cells to the supernatant. The cell-rounding ( visible on the Figure 1E) might be associated with activation of anoikis, also a non-apoptotic type of cell death. This should be discussed ( see https://pubmed.ncbi.nlm.nih.gov/31154235/). I do not see that onco-necrosis has been proved. Anoikis is also possible.
- Increased level of TNFα mRNA levels indicates activation of synthesis. How is this possible considering such a strong release of ATP via permeabilization? Discussion section should address these contradictions.
- Figure 6 looks incomplete. You need to indicate what happens when DC maturation is complete; DC can kill cancer cells etc. Cell death should be replaced with anoikis/necrosis etc.
- Limitations of this work were poorly presented. You need to indicate that the role of DCS requires further investigation as in some cases DC can promote cell migration and metastasis etc.
Author Response
SUBJECT: Response Letter to Reviewer 1
Dear Reviewer,
We appreciate helpful suggestions and constructive comments provided by the reviewer. We thank the reviewer for finding the paper interesting and well written. We have revised the manuscript in response to the reviewers’ comments and believe that our revisions have significantly improved the quality of this manuscript. Below please find our point-by-point response to the comments of the reviewer, as shown in blue arial font.
Reviewer #1:
Comment: Authors discovered a novel compound, thieno-pyrimidine-derived molecule TPH104, which activates cell death in the Triple negative breast cancer cells (TNBC) via complex effectors including dendritic cells (DC) and pro-inflammatory cytokine production. The paper is well written and interesting. Although several cell types were used to address the research question, only one TNBC cell line was actually used to prove the effect of TPH104 (MDA-MB-231--- this is a serious limitation of this study).
Response: We thank you for your encouraging words about the manuscript. In the present study, we used MDA-MB-231 cells to assess the cell death phenomena and related immunogenic effects of a novel compound TPH104. We have carried out extensive studies to understand the cell death mechanism of TPH104 in a range of TNBC cell lines including BT20, SUM159PT and MDA-MB-468. These studies are a part of another manuscript where we show extensive chemistry of TPH class of compounds on TNBC cells, their efficacy, safety, anti-metastatic properties and non-apoptotic cell death mechanisms. The present study is intended to be a short communication about the interesting observation that shows immunogenic nature of cell death induced by TPH104 in TNBC. In view of this, we chose MDA-MB-231 cells as an experimental model of TNBC to test our hypothesis of immunogenic cell death by TPH104 and performed relevant experiments accordingly. Nevertheless, we have shown additional data on another TNBC cell line BT-20. Thank you.
Comment: 1. “TPH104 (5 µM)” – the dose is quite high and might be nonspecific. Authors should indicate what vehicle was used and discuss a possible Vehicle and TPH interactions.
Response: We thank you for your comment. We used DMSO at a concentration of <0.1% as vehicle to deliver TPH104 in our culture systems. We do not know of any interactions of TPH104 and DMSO. The higher concentration of TPH104 at 5uM was used for a shorter duration study to capture the mechanism of cell death. Alternatively, we could have used a smaller concentration for a longer duration, however, that will make it difficult to capture the precise cell death process.
Comment: 2. It is contradictory that TPH104 stimulated both non-apoptotic cell death and increased ATP levels from cancer cells to the supernatant. The cell-rounding (visible on the Figure 1E) might be associated with activation of anoikis, also a non-apoptotic type of cell death. This should be discussed (see https://pubmed.ncbi.nlm.nih.gov/31154235/). I do not see that onco-necrosis has been proved. Anoikis is also possible.
Response: We thank you for raising this critical concern. We would like to point out that the precise molecular mechanisms that induce secretion of ATP during ICD is not clear. We understand that considering the requirement of apoptotic machinery (caspase3 dependent cleavage of pannexin1 channels for ATP efflux during ICD) (PMID: 23852373) and autophagy-competency of cancer (PMID: 22174255) for active secretion of ATP during ICD might contradict our findings pertaining to ATP release during a non-apoptotic cell death induced by TPH104 in an autophagy incompetent TNBC (PMID: 32732922) However, there is a possibility of caspase/pannexin-1 independent mechanism of ATP release (PMID: 25112874) that prompted us to appreciate our findings in a non-apoptotic mode of cell death by TPH104. We further thank you for prompting us towards the possibility of anoikis like cell death induced by TPH104 in MDA-MB-231 cells. The intrinsic pathway of anoikis depends on caspase activation and mitochondrial permeabilization (PMID: 16247493; PMID: 31154235). Furthermore, anoikis features exhibit phosphatidylserine externalization like apoptosis. Our unpublished studies indicate no activation of caspases or annexin staining. In view of this, it seems unlikely that cells are undergoing apoptotic cell death. In addition, during apoptotic cell death protein synthesis is actively abolished through caspase-mediated destruction of translational machinery of cells. whereas in TPH104 induced cell death corpse of MDA-MB-231 still produce TNFα indicating intact ER network, which is also a feature of non-apoptotic cell death. Further experiments are underway to address the mechanism of cell death operated by TPH104 in TNBCs.
Comment: 3. Increased level of TNFα mRNA levels indicates activation of synthesis. How is this possible considering such a strong release of ATP via permeabilization? Discussion section should address these contradictions.
Response: We thank you for raising a truly relevant question pertaining to TNFα expression. There are many recent reports that indicate that activation of necroptotic pathway is associated with inflammatory transcription and cell death simultaneously (PMID: 31227597; PMID: 26405229). Further, it has been reported that unlike apoptotic cell death (during which the protein synthesis is actively abolished through caspase-mediated destruction of translational machinery of cells), necroptotic corpse of cells can continue to produce protein via intact ER network even after loss of plasma membrane integrity (PMCID: PMC6857709). These cell death mechanisms are extraordinarily complex and need more thorough understanding. At this point based on our observation of active transcription of TNF and corresponding proteins in the culture supernatants further substantiate that TPH104 drive a non-apoptotic cell death in MDA-MB-231 cells that needs extensive exploration.
Comment: Figure 6 looks incomplete. You need to indicate what happens when DC maturation is complete; DC can kill cancer cells etc. Cell death should be replaced with anoikis/necrosis etc.
Response: Thank you. We have made revisions as suggested.
Comment: English language level should be improved; 2-abbreviation Hie should be His?
Response: We thank you for pointing out the lacunae in figure 6. We have made appropriate changes to complete the figure.
Comment: 5. Limitations of this work were poorly presented. You need to indicate that the role of DCS requires further investigation as in some cases DC can promote cell migration and metastasis etc.
Response: We thank you for your comments. We have now discussed the limitations of the work i.e.
1.Lacks clarity with regards to cell death mechanism.; 2.We have investigated the immunogenic cell death by TPH104 in a single MDA-MB-231 cell.; 3. Lacks mechanistic insight to our observation of TPH104 induced TNF α production by MDA-MB-231 cells.; 4.Needs further substantial studies.
Thank you again for critically reviewing our work and providing constructive comments. We have further revised our manuscript. I hope you will find our revised manuscript acceptable for publication in Cancer’s journal. I thank you for your consideration.
Sincerely yours,
Amit K. Tiwari, Ph.D.

Reviewer 2 Report
The manuscript submitted by Tukaramrao et al. reports a novel in-house synthesized compound, TPH104, which can induce cell death and immunogenic response in the TNBC MDA-MB-231 cell line. In addition, the authors connect the immunogenic cell death of the cell line with the dendritic cells (DC) activation in vitro. There are several concerns from the reviewer as listed below.
- The introduction is too long, and many contents are unrelated to the main conclusion of this study. e.g., the paragraph from Line 80-92, there is no data related to this topic. Consider shortening, removing, or moving to the discussion part. Some essential knowledge should be introduced, e.g., Line 394, the CD11C+ population of the DC should be described in the introduction.
- The cytokine expressions from BMDCs which inducted by the conditioned media from TPH104 treated MDA-MB-231 cells are very weak. According to Figure 5, only the higher concentration TPH104-5-CM can induce a mild fold change of mRNA expression for some of the cytokine. On the contrary, the induction of TNFa in MDA-MB-231 by TPH104 treatment was dramatic. What will be the difference in terms of TNFa induction either by TPH104 treated MDA-MB-231 or activated DC in the physiological condition? Although in the in vitro condition, TPH104 treated MDA-MB-231 causes DC activation, the phenomenon will be consistent in vivo condition is unknown and there is a lack of evidence.
- What's the rationale to use the gingival keratinocytes as control cell rather than human mammary epithelial cells such as MCF-10A?
- From figure 1D, it's hard to tell that TPH104 produced a concentration-dependent decrease in cell confluence, the three concentrations caused a very similar cell proliferation suppression rate. However, in figure 1C, it seems like the cell density at 0h is unequal in different treatment groups. Does the value shown in 1D normalized by the time 0h?
- Line 310-312, data was not showed. Again, this experiment should conduct using a normal mammary epithelial cell line.
- Add quantification of the CRT induced by the treatments as shown in Figure 2B.
- Line 359-360. The sentence "compared to control the cells incubated..." should be rewritten. Any idea that TPH104 treatment at 12h caused some cell proliferation inhibition (Figure 1D) while no TNF induction (Figure 3A)?
- Add the purpose of using the LPS treatment experiment as described in Line 403-405, Figure 4E.
- There are several mistakes in the figure legends of Figure 4. e.g., the tile of the panel (B) is missing, tile for the CD86 population should be (D) according to the figure. There are inconsistent statements regarding the conditioned media collection time in the text (Line 393), and in the figure legend (Line 417)
- There are some suggestions to improve the figures. 1) Add gene name on top of each panel in Figure 5; 2) Statistical markers were missing in Figure 5B; 3) The y-axis font in Figure 3A and Figure 5 need to be improved; 4) Figure 6 was not fully showed.
- There are some mistakes that should be correct throughout the manuscript. e.g.,
Line 59, this abbreviation already showed in line 45.
Line 409, MHCHi (p<0.5) or p<0.05?
Author Response
SUBJECT: Response Letter to – Reviewer 2
Dear Reviewer,
We appreciate helpful suggestions and constructive comments provided by the reviewer. We thank the reviewer for finding the paper interesting and well written. We have revised the manuscript in response to the reviewers’ comments and believe that our revisions have significantly improved the quality of this manuscript. Below please find our point-by-point response to the comments of the reviewer, as shown in blue arial font.
Reviewer #2:
Comment: The manuscript submitted by Tukaramrao et al. reports a novel inhouse synthesized compound, TPH104, which can induce cell death and immunogenic response in the TNBC MDA-MB-231 cell line. In addition, the authors connect the immunogenic cell death of the cell line with the dendritic cells (DC) activation in vitro. There are several concerns from the reviewer as listed below.
Response: We thank you for thoughtful comments. We have revised our manuscript accordingly.
Comment: The introduction is too long, and many contents are unrelated to the main conclusion of this study. e.g., the paragraph from Line 80-92, there is no data related to this topic. Consider shortening, removing, or moving to the discussion part. Some essential knowledge should be introduced, e.g., Line 394, the CD11C+ population of the DC should be described in the introduction.
Response: We thank you for your comment. We thank you for pointing out the discontinuity in the flow of information in the introduction. We have removed the paragraph between line 80-92. As suggested a paragraph describing the importance of CD11C+ DCs in triple TNBC has been incorporated -- Tumor infiltrating lymphocytes and tertiary lymphoid structures have favorable prognostic value in TNBC (PMID: 29848327). The immune response in tumors is mainly depend on adaptive immunity, mediated by T cells (PMID: 25634101). CD8+ T cells evolve and kill tumor cells by exerting perforin, granzymes and INF-γ. (PMID: 22190859). Dendritic cells (DCs) are central to the initiation of primary immune responses and they are the only antigen-presenting cells capable of stimulating T cells, making them pivotal in the generation of adaptive immunity (PMID: 29848327). In triple negative breast cancer patients with higher levels of tumor-infiltrating lymphocytes (TIL) and tertiary lymphoid structures (TLSs) in the TME have better outcomes (PMID: 28824872; PMID: 26475777). And recent reports suggest that CD11c+ dendritic cells have a strong correlation with TILs and TLSs in TNBC (PMID: 30348717). Therefore, therapeutic modalities that facilitate activation and recruitment of CD11+ DCs find value.
Comment: 2. The cytokine expressions from BMDCs which inducted by the conditioned media from TPH104 treated MDA-MB-231 cells are very weak. According to Figure 5, only the higher concentration TPH104-5-CM can induce a mild fold change of mRNA expression for some of the cytokine. On the contrary, the induction of TNFa in MDA-MB-231 by TPH104 treatment was dramatic. What will be the difference in terms of TNFa induction either by TPH104 treated MDA-MB-231 or activated DC in the physiological condition? Although in the in vitro condition, TPH104 treated MDA-MB-231 causes DC activation, the phenomenon will be consistent in vivo condition is unknown and there is a lack of evidence.
Response: We thank you for pointing out the mild or no changes in cytokine expression caused by factors in conditioned media of TPH104 (1 µM) treated MDA-MB-231 cells and accept this critique. As pointed out, TPH104 at 5µM levels induced significant increase in expression of TNFα and IL-12A. TPH104 at 1-5 µM was effective in inducing maturation of dendritic cells (elevated expression of CD86, MHCII; Fig. 4) and T cell activation potential suggested by the expression of IL-12A (Fig. 5D). However, the activation of TNFα expression in DCs was more pronounced by 5 µM of TPH104 CM of MDA-MB-231 cells. This could be due to significant increase in levels of TNFα in 5 µM of TPH104 CM of MDA-MB-231 cells (Fig.3B). Moreover, it is known that TNFα is a weak inducer of DC maturation comparison to other DAMPS (PMID: 11086063). Therefore, it seems that the CM of 5 µM TPH104 CM of MDA-MB-231 cells could have other uncharacterized DAMPS that might be contributing to increased levels of TNFα transcripts in DCs.
The second question pertaining to the difference in terms of TNFα induction between the CM of TPH104 treated MDA-MB-231 cells and activated DC in the physiological condition – bone marrow derived DCs in the present study were grown in media containing recombinant mouse GM-CSF and can be considered as closer to physiological condition and the level of TNFα produced under these conditions by DCs was not measured in the present study. But from published literature (PMID: 11086063) it is evident that adding 20ng/mL of TNFα to the BMDCs cultured with GM-CSF and IL-4 will bring about a mild increase in the expression of MHCII and CD86.
Comment: 3. What's the rationale to use the gingival keratinocytes as control cell rather than human mammary epithelial cells such as MCF10A?
Response: We thank you for your comments. We wanted to show the safety and selectivity of TPH104 in cancer cells compared to other cell lines. We have now removed gingival keratinocytes data as controls. Instead, we have included BT-20 cell line data as requested by other reviewer. We tried to recover MCF10A and HMEC normal epithelial cells but due to short timeline provided to us for revising this manuscript we were unable to submit those data. Nevertheless, we have tested many other cancer cells and TPH104 is pretty specific to TNBC cells (data not shown). Thank you.
Comment: 4. From figure 1D, it's hard to tell that TPH104 produced a concentration-dependent decrease in cell confluence, the three concentrations caused a very similar cell proliferation suppression rate. However, in figure 1C, it seems like the cell density at 0h is unequal in different treatment groups. Does the value shown in 1D normalized by the time 0h?
Response: We thank you for your keen observation and pointing out the finer details in figure1. We have made appropriate changes.
Comment: 5. Line 310-312, data was not showed. Again, this experiment should conduct using a normal mammary epithelial cell line.
Response: Thank you for your comments. We have removed the sentence between line 310- 312. Please see our response for comment 3. Thank you.
Comment: 6. Add quantification of the CRT induced by the treatments as shown in Figure 2B.
Response: Thank you for your comment. The quantitation of CRT is added to figure 2B.
Comment: 7. Line 359-360. The sentence "compared to control the cells incubated..." should be rewritten. Any idea that TPH104 treatment at 12h caused some cell proliferation inhibition (Figure 1D) while no TNF induction (Figure 3A)?
Response: Thank you for your observation. We have corrected the sentence in line 359-360.
Twelve hours post TPH104 treatment of MDA-MB-231 cells there is inhibition of proliferation but at that time point we did not observe any change in the mRNA levels of TNFα. So it is imperative that TNFα production in these cells began between 12- 24h post TPH104 treatment of MDA-MB-231 cells. Data in fig. 1C, showing no effect of TNFα neutralizing antibody on inhibition of proliferation caused by TPH104 MDA-MB-231 cells measured after 48h clarifies that TNF α is not involved in driving the MDA-MB-231 cell death but it is a byproduct of TPH104 induced cell death. Furthermore, there are many recent reports that indicate that activation of necroptotic pathway is associated with inflammatory transcription and cell death simultaneously (PMID: 31227597; PMID: 26405229). This point is elaborated in response to comment 3 of reviewer1.
Comment: 8. Add the purpose of using the LPS treatment experiment as described in Line 403-405, Figure 4E.
Response: Thank you for your comment. Assessment of BMDC activation by 100ng/mL LPS was performed as a positive quality control experiment to assess the optimal activation of BMDCs. This sentence has been added.
Comment: 9. There are several mistakes in the figure legends of Figure 4. e.g., the tile of the panel (B) is missing, tile for the CD86 population should be (D) according to the figure. There are inconsistent statements regarding the conditioned media collection time in the text (Line 393), and in the figure legend (Line 417).
Response: Thank you for pointing out the inconsistencies pertaining to conditioned media collection and details in the legend of figure 4. Appropriate changes have been made respectively.
Comment: 10.There are some suggestions to improve the figures. 1) Add gene name on top of each panel in Figure 5; 2) Statistical markers were missing in Figure 5B; 3) The y-axis font in Figure 3A and Figure 5 need to be improved; 4) Figure 6 was not fully showed.
Response: We value your suggestions. 1) Gene names have been placed on top of each graph in Fig. 5, 2) missing statistical mark in Figure 5B has been added, 3) The y-axis font in Figure 3A and Figure 5 have been improved. 4) Figure 6 has been updated
Comment: 11. There are some mistakes that should be correct throughout the manuscript. e.g., Line 59, this abbreviation already showed in line 45. Line 409, MHCHi (p<0.05?
Response: We thank you for your comments. These mistakes have been corrected.
Thank you again for critically reviewing our work and providing constructive comments. We have further revised our manuscript. I hope you will find our revised manuscript acceptable for publication in Cancer’s journal. I thank you for your consideration.
Sincerely yours,
Amit K. Tiwari, Ph.D.

Round 2
Reviewer 1 Report
I am satisfied with the current revised version of the manuscript. The authors addressed all my comments /suggestions properly.
Author Response
Thank you!
Reviewer 2 Report
The authors have addressed all the comments from the reviewer, the revised manuscript has been largely improved. The only concern is the toxicity and selectivity of the compound, because there is no data from treatment in other cell lines in the revised version. The authors responded they have tested other cell lines which could suggest to acknowledge the unshown data and make further discussion.
Author Response
Thank you for approving all other comments. We again thank you for re-reading our manuscript and finding it suitable for the most part. Thank you for raising question on selectivity and safety of these compounds. As mentioned earlier, the ideal situation would be to use normal breast epithelial cells to measure the selectivity. However, our HMEC and MCF10A cells are/were not co-operating. And due to limited time provided for revision it is difficult to get these cells working. We however, redid our analysis on the normal fibroblast cells CRL1459 cells and also decided to re-insert the normal gingival epithelial cells data to compare and contrast safety and selectivity of TPH104 compounds. We have revised our data in figure 1. I hope this will be OK.
An exhaustive toxicity studies on TPH104 was also performed recently and all the safety data is being compiled and is being presented in another manuscript.
We welcome Reviewer further advise.